# Seroma after Simple Mastectomy in Breast Cancer—The Role of CD4+ T Helper Cells and the Evidence as a Possible Specific Immune Process

**DOI:** 10.3390/ijms23094848

**Published:** 2022-04-27

**Authors:** Nicole Pochert, Mariella Schneider, Nadine Ansorge, Annamarie Strieder, Jacqueline Sagasser, Matthias Reiger, Claudia Traidl-Hoffmann, Avidan Neumann, Udo Jeschke, Christian Dannecker, Thorsten Kühn, Nina Ditsch

**Affiliations:** 1Department of Obstetrics and Gynecology, University Hospital Augsburg, Stenglinstrasse 2, 86156 Augsburg, Germany; nicole.pochert@med.uni-augsburg.de (N.P.); mariella.schneider@uk-augsburg.de (M.S.); nadine.ansorge@uk-augsburg.de (N.A.); annamarie.strieder@uk-augsburg.de (A.S.); jacqueline.sagasser@uk-augsburg.de (J.S.); christian.dannecker@med.uni-augsburg.de (C.D.); nina.ditsch@uk-augsburg.de (N.D.); 2Department for Environmental Medicine, University Hospital Augsburg, Neusässer Straße 47, 86156 Augsburg, Germany; matthias.reiger@med.uni-augsburg.de (M.R.); claudia.traidl-hoffmann@med.uni-augsburg.de (C.T.-H.); avidan.neumann@tum.de (A.N.); 3Department of Obstetrics and Gynecology, Hospital Esslingen, Hirschlandstraße 97, 73730 Esslingen am Neckar, Germany; t.kuehn@klinikum-esslingen.de

**Keywords:** breast cancer, simple mastectomy, seroma formation, lymphocyte enrichment, T-helper cells, Th2 cells, Th17 cells

## Abstract

Seroma development after breast cancer surgery is the most common postoperative complication seen after mastectomy but neither its origin nor its cellular composition is known. To investigate the assumption of immunological significance, one of the first aims of this pilot study is to describe the cellular content of collected seroma fluids and its corresponding serum in patients with simple mastectomy after needle aspiration, as well as the serum of healthy controls. The content of red blood cells (RBC) was measured by haemato-counter analyses, and the lymphocyte identification/quantification was conducted by flow cytometry analyses in seroma fluid (SFl) and the sera of patients (PBp) as well as controls (PBc). Significantly lower numbers of RBCs were measured in SFl. Cytotoxic T cells are significantly reduced in SFl, whereas T helper (Th) cells are significantly enriched compared to PBp. Significantly higher numbers of Th2 cells were found in SFl and PBp compared to PBc. The exact same pattern is seen when analyzing the Th17 subgroup. In conclusion, in contrast to healthy controls, significantly higher Th2 and Th17 cell subgroup-mediated immune responses were measured in seroma formations and were further confirmed in the peripheral blood of breast cancer (including DCIS) patients after simple mastectomy. This could lead to the assumption of a possible immunological cause for the origin of a seroma.

## 1. Introduction

Seromas are one of the most common postoperative complications seen after mastectomy and axillary surgery [1,2,3,4]. In reviews comparing different studies, in 3%, 10%, and up to 85% of all cases, seroma formation occurs mostly within the first weeks after breast surgery [5,6]. The latter finding leads to the possible statement that seroma formation is so common that it could be a side effect of surgery rather than a complication. Nevertheless, seromas in need of needle aspiration have an impact on the patient’s wellbeing and can also significantly impact treatment by delaying adjuvant therapy and increasing the risk of the development of infections after primary surgery [7]. There is evidence that surgical trauma after sentinel lymph node biopsy for breast cancer treatment and seroma development is associated with age in some studies [8]. Several publications discuss surgical procedures to reduce the risk of liquid formation [9,10,11,12]. There is a growing number of studies published in recent years investigating the origin of the fluid under skin flaps or in the axillary dead space following mastectomy and/or axillary dissection. In the 1990s, it was proposed that the fluid is of lymphatic origin, but later studies showed that the cell content of seroma liquid differs from lymph fluid. It has a higher protein content and no fibrinogen present, making coagulation impossible [13]. After further investigation, the fluid appeared to be composed more similarly to an exudate resulting from an inflammatory reaction during the first phase of wound repair [14]. In contrast to this preliminary data, a study by Monalto et al. showed by analyzing cell content and cytokine analysis that seroma fluid derives from an accumulation of afferent lymph [15]. 

Regarding our assumption of an immunological connection to the development of a seroma, this work investigates the special subgroups of immune cells that are part of the immunomodulatory process and will therefore be presented in more detail. Immunity is divided into innate and adaptive immune responses. The former reacts rapidly and non-specifically to pathogens, whereas the latter responds in a slower but more specific manner with the generation of long-lived immunological memory [16]. Innate immunity is mediated by innate immune cell populations such as myeloid cells, natural killer (NK) cells, and innate lymphoid cells, as well as by ancient humoral systems such as defensins, and complement them as the first line of defense against infections within minutes to hours [17,18]. Adaptive immunity is orchestrated by CD4+ T helper cells (Th) via the production of cytokines and chemokines that enhance cytotoxic CD8+ T cell responses and are indispensably required for B cell-dependent antibody production and plasma cell generation [19,20]. Th cells are divided into regulatory T cells (Treg) and effector T cells. These effector T cells can be subdivided into several types according to their cytokine secretion profile [21]. The diverse functions of CD4+ Th cells are determined by their cytokine secretion patterns and display a large degree of plasticity and the ability to differentiate into multiple sublineages in response to developmental and environmental cues [22,23]. These differentiated sublineages can orchestrate a broad range of effector activities during the initiation, expansion, and memory phase of an immune response as well as regulatory activities. Emerging evidence suggests that Th cells also actively participate in shaping antitumor immunity [23].

The controversial discussion about seroma formation and composition shows that the underlying mechanisms of seroma development remain unclear [24,25]. Based on the lack of scientific knowledge about seromas, there is a need to understand the underlying mechanisms of seroma formation for possible prevention and/or better treatment strategies. 

Therefore, one of the aims of this pilot study is to describe the cellular content of collected seroma fluids in patients with simple mastectomy after needle aspiration to elucidate the type of specific immune reaction. Secondarily, we focus on analyzing the distribution of different immune cell subsets found in seroma liquids as well as of peripheral blood of the same patient after mastectomy in comparison with the peripheral blood of healthy volunteers. 

## 2. Results

### 2.1. Patient Cohort

For the evaluation of the above-mentioned suggestions, a subset of all these patients fulfilling the criteria of the study and who presented with breast cancer (n = 16, two with contralateral disease), simple mastectomy, and consecutive seroma formation between 10/2020 and 09/2021 was chosen. Personal characteristics of both groups and tumor characteristics of the study population are described in Table 1 and Appendix A.

Appendix A include information about the therapeutic approaches of the study group as well as the seroma aspirations including time points and the volume of seroma fluid (SFl). Patients who received chemotherapy before surgery and an axilla dissection tended to develop more seromas. Patients who received chemotherapy before surgery develop seromas at equal frequencies compared to patients without chemotherapy. Moreover, the development of seromas with a volume of more than 100 mL seems to not be influenced by the type of surgery in the axilla. Patients without seroma development were excluded from this study since we could not see any significant changes in their peripheral blood composition and our sample size was too small (Appendix A).

### 2.2. Basic Cell Composition in Seroma Fluids

Upon analyzing red blood cell (RBC) content via the haemato-counter, we found significantly reduced numbers of RBCs within the aspirated SFl in comparison to the peripheral blood of patients (PBp) and controls (PBc) (Figure 1A). The mean abundance of RBCs was 60.9 mil cells per mL SFl, whereas the average number of RBCs in PBp is 3943 mil cells/ml and 4529 mil cells/ml for PBc. The difference between PBp and PBc is also highly significant. We analyzed the data acquired when staining the cells of SFl and PBp/c with antibodies included in panel 1 for the expression of the pan-leucocyte marker CD45 (Figure 1B). 

Almost all of the SFl showed an abundance of leukocytes higher than 90% of all recorded events. One sample had a content of 76.25% leukocytes and one with only 0.2%. The latter sample was excluded from further analysis because no immune cells could be measured. The gating strategy of panel 1 is displayed in Figure 1C. Based on the location of the CD45+ leucocytes within the side scatter (SSC) vs. forward scatter (FSC) [26] plot, we determined the percentages of granulocyte, monocyte, and lymphocyte subpopulations (Figure 1D). It is important to note that granulocytes as well as monocytes in SFl did show a rather heterogeneous character, as well as not so clear cells populations as compared to PBp/c in the SSC vs. FSC plot. This prompted us to draw gates for granulocytes, as well as monocytes, slightly bigger to be able to include all cells of the particular cell type. 

SFl showed significantly reduced numbers of granulocytes with a mean of 20% of all leucocytes compared to PBp with 62% and PBc with 60% granulocytes within the leucocyte compartment. The same pattern was seen within the monocyte subpopulation. In PBp/c, the average percentage was approximately 8% of all leucocytes compared to SFl with 4.6%. In contrast, the lymphocyte compartment had the highest abundance in SFl with an average of 63% of all CD45+ cells compared to approximately 27% in PBp/c. 

### 2.3. Lymphocyte Composition in Seroma Liquid and PB Analyzed by Flow Cytometry

Since lymphocytes were the dominant population within SFl, further examination was focused on analyzing the flow cytometry data of panel 1 described in Table 2 and Figure 2A in regards to the expression of the T cell marker CD3, the B cell marker CD19, as well as CD56 and CD16 as markers for natural killer (NK) cells. We found a significantly enhanced CD3+ T cell population in SFl (87.2 ± 7.3%) compared to PBp (71.6 ± 8%) as well as PBc (75.2 ± 4.8%) (Figure 2B). 

On the other hand, the CD19+ B cell population was diminished in almost all SFls (5.5 ± 5.7%) compared to PBp (13 ± 5.6) and PBc (12.7 ± 3%). The same pattern is seen for NK cells. SFl showed significantly reduced percentages of NK cells (4.3 ± 2.2%) compared to PB in general. As a result, PBp had significantly higher numbers of NK cells (15.9 ± 7.8%) compared to PBc (11.3 ± 4.2%). The numbers of peripheral blood in both study participants and control subjects were within the normal range of healthy individuals between 55 and 83% T cells, 6 and 19% B cells, and 7 and 31% NK cells within the lymphocyte population [27]. We used these reference values in particular since those are obtained similarly to our gating strategy.

In contrast to the positive results for the marker CD3 (NKT cells) in the PBc/p populations, there were significantly fewer cells in SFl, except in three seroma samples with relatively high NKT levels (Figure 2C). Almost all of the measurements were within the reference values for NKT cells of 1–18% of all lymphocytes [28]. Upon examining the activation marker HLA-DR, we found a significantly higher number in SFl (36.1 ± 12%) compared to PBp (24.3 ± 10.3%) and PBc (17.6 ± 5.2%). (Figure 2C). 

### 2.4. T-Cell Identification

#### 2.4.1. Cytotoxic vs. T-Helper Cell Analysis

Since T cells (CD3+) were the most frequent leucocyte population within SFl, data were analyzed in regards to their CD4 and CD8 expression. Cytotoxic T cells, positive for the specific marker CD8, were significantly reduced in SFl, whereas CD4+ Th cells were significantly higher (CD8+: 14.2 ± 6.7%; CD4+: 82.8 ± 7.6%) compared to PBp (CD8+: 25.2 ± 8.9%; CD4+: 69.5 ± 10.7%) and PBc (CD8+: 26.9 ± 9%; CD4+: 67.2 ± 11.4%) (Figure 3A,B). 

#### 2.4.2. T Helper Cell Subpopulation Analysis

To analyze effector T helper cell subpopulations, we adapted the gating strategy as described by an isolation protocol provided by Miltenyi Biotec (panel 2, Table 2) (Miltenyi Biotec, Bergisch Gladbach, Germany) [29]. We were able to determine the Th1, 2, 9, 17, and 22 subpopulations within the CD4+ T helper cell compartment for SFl, as well as PBp/c, based on their lineage-specific chemokine receptor profiles (Figure 3C and Appendix A). The analysis of the flow cytometry data revealed significantly higher percentages of Th2 cells gated on CD4+ lymphocytes in SFl (7.2 ± 2.5%) and PBp (6.6 ± 2.3%) compared to PBc (3.6 ± 1.6%), whereas SFl and PBp did not show significant differences. The exact same pattern is seen within the Th17 subgroup. In addition, the number of Th22 cells is significantly higher in SFl (11.3 ± 7.7%) compared to PB, but in contrast to Th2 and Th17, there were also significant differences between SFI and PBp (3.5 ± 1.2%) and not between PBp and PBc (2.9 ± 0.8%). Th1 and Th9 subgroups did not show any significant difference between SFl and PBp/c. 

#### 2.4.3. Treg Subpopulation 

To investigate regulatory T (Treg) cells, we gated the surface marker CD25+/CD127-/low cells within the CD4+ T helper cell population (panel 3; Figure 3D and Table 2) [30,31]. By analyzing the data, we found no significant differences in Treg numbers when comparing SFl, PBp, and PBc (Figure 3E).

We used CD45RA as a marker of cell activation. It is a fact that CD45RA+ T cells are converted to CD45RO+ T cells after activation [32]. Within this naïve CD45RA+ Treg population, significant differences were detected between naïve Tregs in SFl (with a mean of 2.2%) and PBc (4.5%). PBp values did not show significant differences compared to either SFl or PBc, with a difference of 3.9% between both study groups. Interestingly, by comparing the numbers of naïve T helper cells in general, there were significantly elevated levels of those naïve cells in PBc (56.7 ± 10%) compared to SFl (20.2 ± 20.1%) and PBp (31.2 ± 7.6%) (Figure 3F).

### 2.5. Differences in Cell Distribution between First and Second Aspiration

Upon comparing the abundance of all leucocyte subtypes between the first and second fine needle aspiration (n = 5), we could not see any significant differences between any given subgroup (Figure 4A and Appendix A).

## 3. Discussion

The aim of this evaluation was the characterization of immunological cell populations in seroma fluid after a simple mastectomy procedure for breast cancer patients. Flow cytometry was used as the analysis method of SFl after needle aspiration. To confirm the nature of the fluid and evaluate possible contamination with PB during the surgical procedure, the number of RBCs within the aspirated fluid was determined and compared with those of PBp/c. There were significantly lower numbers of RBCs in SFl compared to PBp/c, leading to the assumption that the SFl formation was not due to a hemorrhage from the blood vessels after surgery. Although RBC numbers of almost all patients and healthy controls were within the range of common reference values of PB, a significant decrease in RBC numbers in patients compared to healthy controls could be seen [33]. The high percentage of CD45+ cells suggests that the cellular fraction of the SFl consists mainly of leucocytes. Nevertheless, the flow cytometry data showed a different distribution of granulocytes, monocytes, and lymphocytes within this leucocyte compartment in SFl compared to PBp and PBc. Similar data were shown in a publication by Montalto et al. in 2010. In agreement with our findings, the authors showed an increased percentage of lymphocytes, as well as lower numbers of granulocytes and monocytes. This indicates a specific adaptive immune response leading to the recruitment of specific lymphocytes to the side of the injury. Instead of monocytes, Monalto et al. found cells with more heterogeneous physical characteristics and only partially expressing CD14. Since we did not stain for any monocyte markers, we cannot prove the latter statement. Nevertheless, we also saw this more heterogeneous population in the SSC vs. FSC plot. These cells might be particularly interesting for further investigations. 

Most of the lymphocytes in SFl were CD3+ T cells and only very low percentages of B cells and NK cells were present. However, PB of the same study participants and healthy controls exhibited normal distributions of those cell subpopulations in comparison to reference values, found in previous literature. Comans-Bitter et al. described reference values for human lymphocyte subpopulations. In agreement with our results, they found a frequency of CD3+ T-cells of 55–83% and CD3/CD4+ Th-cells of 28–57%. The near absence of NK further supports the hypothesis of a specific adaptive rather than innate immune response due to bacterial contamination during surgery. 

Within the T cell compartment, CD4+ T helper cells rather than CD8+ cytotoxic T cells were identified in seroma fluids. Th2, Th17, and Th22 were highly abundant in SFl, whereas the number of Tregs showed no significant difference. These findings indicate a high inflammatory proceeding without changes in the protective regulatory responses through Tregs. There are significantly lower numbers of naïve cells (Th and Treg) present in patients, in the SFl as well as PBp, implicating a high activation status of these immune cells. The significantly lower fraction of naïve CD4+ Th cells in the blood of the patient cohort compared to healthy individuals could potentially be a biomarker for assessing the risk of patients for seroma development. 

Cells positive for the activation marker HLA-DR showed a significantly higher abundance in SFl, which supports our hypothesis of an active inflammatory response preceding at the side of the immune response. This was different from blood, where there is a high proportion of non-activated cells present. HLA-DR+ T helper cells were shown to be increased in patients with active tuberculosis. Furthermore, Tippalagama et al. demonstrated that proliferating CD4+ effector T cells express high levels of HLA-DR. These findings suggest that HLA-DR might be a useful marker for monitoring immune responses [34]. 

One limitation of this study could be the fact that some of our findings, especially differences between PBp and PBc, might be explained by the difference in the average age between the study cohort and control group. It has been shown that NK cell frequency increases with age, whereas naïve T cell numbers decline [35,36,37]. On the other hand, Th17 cells also decline with age, but in our case, Th17 cells are significantly higher in the blood of the older study participant group compared to the younger control group [38]. However, SFl and PBp were taken from the study participants at the same time point. which was the basis of this study.

There is evidence that these WBCs are actively recruited to the side of infection/operation regarding wound healing [39]. Besler et al. [39] described that seroma formation is due to inflammatory mediators’ migration to the side of the tissue trauma. In addition, seroma formation may also be due to a prolonged inflammatory phase due to the disruption of certain stages of wound healing [39]. Th22 cells are believed to play important roles both in promoting wound healing and the repair of damaged epithelial barriers as well as in enhancing immune responses against some pathogens [21,40,41]. The correlation between Th22 cell action and wound healing could in part explain the high numbers of Th22 cells in SFl in most of our patients. 

Supporting our hypothesis, recent studies have shown that there is a high number of dendritic cells found in SFl, which are known to efficiently stimulate the T cell response by recruiting T helper cells [42]. Concerning breast cancer, we are just getting started in our understanding of the role of Th17/Th22 cells, whereas for other tumor types, both cell populations seem to be involved in inflammatory processes leading to cancer formation, as well as in new immune-related cancer treatment approaches [43]. The absence of Th1 cells suggests that there is no evidence of bacterial infections [44]. Th1 cells secrete a number of specific cytokines, which allow these cells to effectively fight infections by viruses and bacteria [45].

To dissect patients at high risk for seroma development, this pilot study serves as a base for the investigation of a high number of PB samples before and after seroma development. In addition, we have to rule out the possibility of bacterial contamination in those cases of severe seromas by analyzing the SFl in regards to bacterial contamination. 

This study shows the successful immune cell subpopulation analysis associated with seroma formation after simple mastectomy. Further studies are ongoing to clarify the role of specific T-helper cell immune responses in the formation of seroma after breast cancer surgery also including the differentiation of breast cancer in general and healthy breasts without the formation of a seroma.

## 4. Materials and Methods

### 4.1. Patient Cohort

Between October 2020 and September 2021, in a monocentric subgroup analysis of the pilot phase of the SerMa Study (EUBREAST5), we analyzed a patient population who developed a seroma fluid formation (SFl) of patients with breast cancer or DCIS and underwent mastectomy surgery without any reconstruction procedure. Of those patients, 34 seroma samples were collected using needle aspiration in up to 5 predefined visits (Appendix A). In five patients, more than one aspiration was performed. All the patients gave informed consent, and an ethical committee permitted the study. SFl as well as the peripheral blood of patients and 15 healthy controls were transported to the laboratory and processed within 24 h (Appendix A).

### 4.2. Sample Processing

After centrifugation of the SFl, the supernatant was removed and cells were counted using a haemato-counter (Sysmex XP300; SYSMEX, Norderstedt, Germany). After RBC lysis (VersaLyse; Beckman Coulter, Krefeld, Germany) for 15 min at room temperature and one washing step with PBS, 4 × 106 white blood cells (WBC) were used for flow cytometry analysis. The remaining cells were viably frozen and stored in liquid nitrogen. 

PBp/c anticoagulated with EDTA was analyzed using a haemato-counter. For immune fluorescent staining, 100 µL PBp/c per staining arm was incubated with 1 mL of VersaLyse for 15 min at room temperature and subsequently washed with PBS. The remaining WBCs were stained with antibodies labeled with immune fluorescent dyes. Aliquots of PBp/c were frozen and stored in liquid nitrogen. 

### 4.3. Flow Cytometry Analysis

WBCs of SFl and PBp/c were stained in three different panels with fluorescence-labeled anti-human antibodies for 10 min at room temperature (Table 2). Single color staining and isotype controls were included for the compensation and confirmation of antibody specificity. Stained cells were subsequently fixed with IO Test3 (Beckman Coulter) for 10 min and washed with PBS before recording using the flow cytometer CytoFlex LX (Beckman Coulter). We made sure to record an appropriate number of cells (50,000 lymphocytes per staining). Results were analyzed using Kaluza software (Beckman Coulter) and gated accordingly (Figure 1C, Figure 2A, Figure 3A,D, Appendix A).

### 4.4. Statistical Analysis

The SFl of all needle aspirations was analyzed, and the first measurement of each patient was included in the initial statistical analysis. Cells present in SFl were analyzed for the pan-leucocyte marker CD45. Only measurements with a percentage of CD45+ cells higher than 75% of total events were included. Statistical analysis was performed using GraphPad Prism. Results were analyzed by either a one-way ANOVA or Kruskal–Wallis-test. A value of *p* < 0.05 was considered statistically significant. 

## 5. Conclusions

Breast cancer patients with seroma formation after simple mastectomy showed a CD4+ T helper cell increase in the seroma fluid samples. To our knowledge, this is the first time that subgroups of T helper cells could be identified and characterized in the seroma of breast cancer/DCIS patients surgically treated with a simple mastectomy. Furthermore, a Th2 and/or a Th17 immune response showed corresponding results of sera and seroma in the same patient. These results lead to the assumption that this is the first step in detecting an immune response in both sera and seroma in contrast to healthy persons. Future planned studies of sera from breast cancer patients at the time of primary surgery as well as from those patients who do not develop a seroma after surgery will provide information about whether serum T-helper determination could be used as a prognostic factor for the development of a seroma.

## Figures and Tables

**Figure 1 ijms-23-04848-f001:**
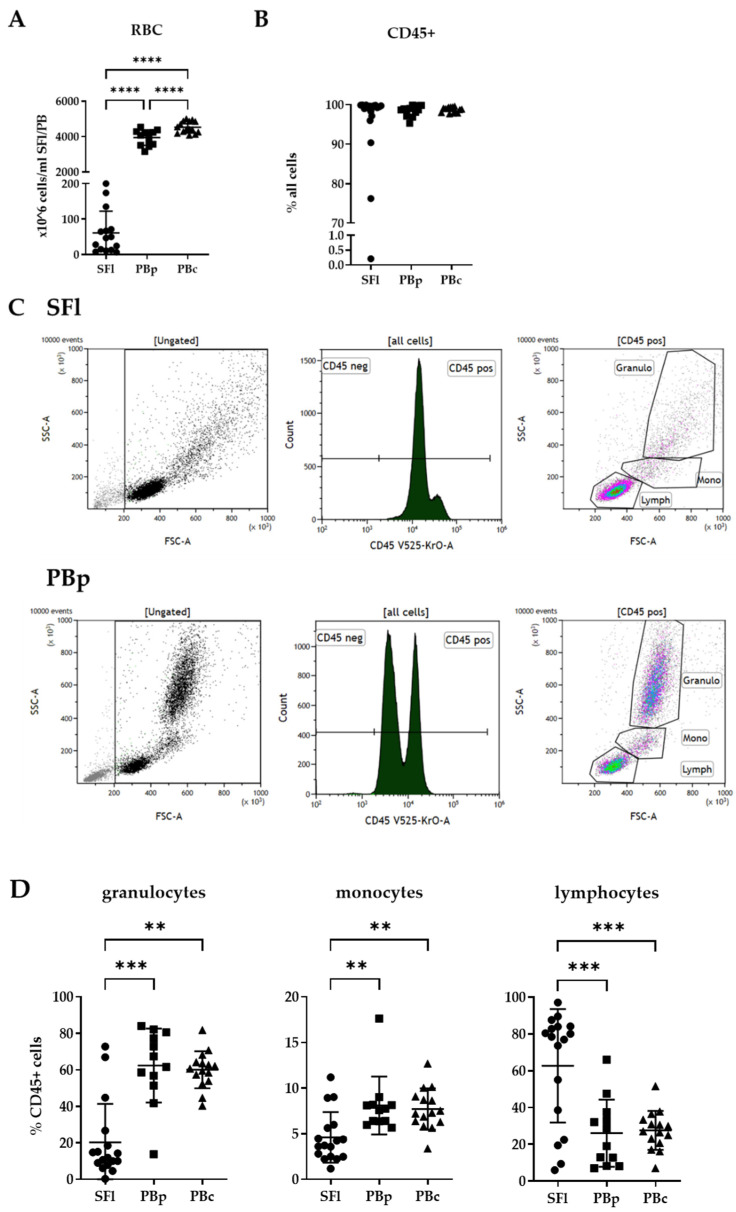
Cell composition of seroma fluid (SFl) and peripheral blood of patients (PBp) and controls (PBc) by flow cytometry analysis or automated cell counting. (**A**) Total cell count of RBCs in SFl and PBp/c determined by automated cell counting (Sysmex). (**B**) Percentage of CD45+ leucocytes in relation to all cells recorded. (**C**) Gating strategy of granulocytes (Granulo), monocytes (Mono), and lymphocytes (Lymph) in SFl and PBp after staining with panel 1; debris was excluded by size discrimination. (**D**) Percentages of granulocytes, monocytes, and lymphocytes within the CD45+ cell population; ** *p* < 0.01, *** *p* < 0.001, **** *p* < 0.0001.

**Figure 2 ijms-23-04848-f002:**
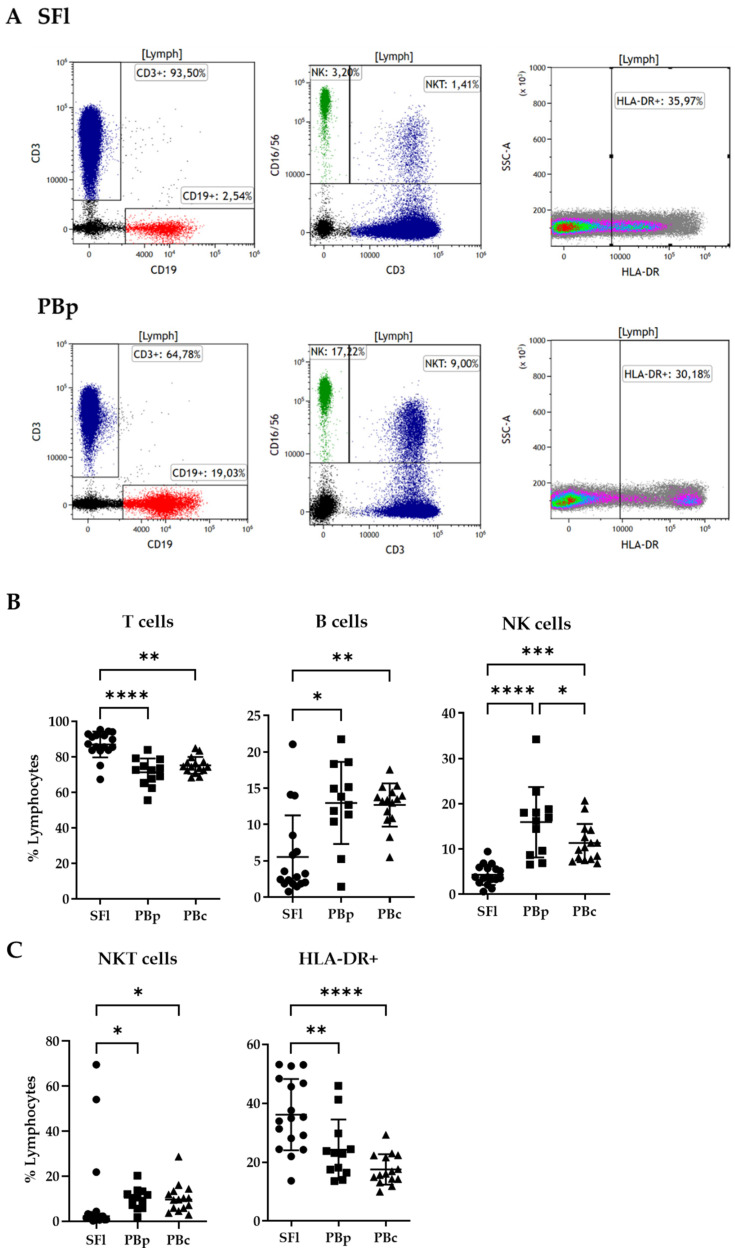
Lymphocyte composition of SFl and PBp/c by flow cytometry analysis. (**A**) Gating strategy of lymphocyte subpopulations in SFl and PBp after staining with panel 1; CD3+ T cells are depicted in blue, CD19+ B cells are shown in red, and NK cells in green. (**B**) Percentage of lymphocytes for CD3+ T cells, CD19+ B cells, and CD56/16+ CD3- NK cells. (**C**) Percentage of lymphocytes for CD56/16+ CD3+ NKT cells and activated lymphocytes (HLA-DR+); * *p* < 0.05, ** *p* < 0.01, *** *p* < 0.001, **** *p* < 0.0001.

**Figure 3 ijms-23-04848-f003:**
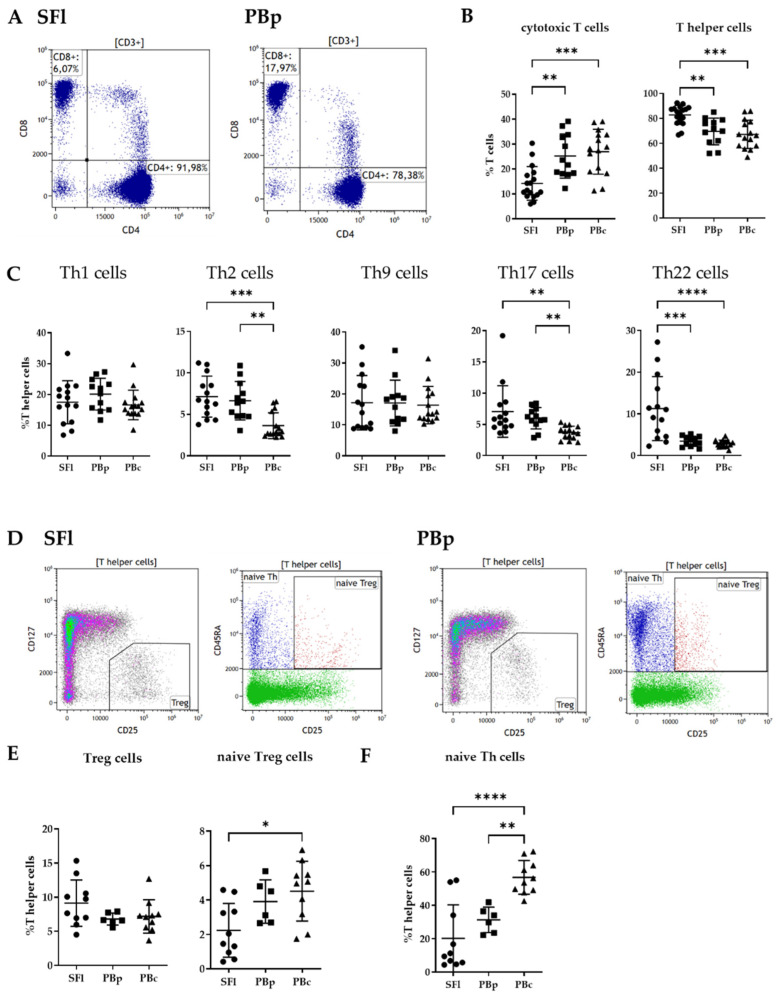
Cell composition of CD3+ T cells in SFl and PBp/c by flow cytometry analysis. (**A**) Gating strategy for the T cell subpopulations CD8+ cytotoxic T cells and CD4+ T helper cells in SFl and PBp after staining with panel 1. (**B**) Distribution of cytotoxic T cells and T helper cells within the T cell compartment. (**C**) Percentage of effector T helper cells within the CD4+ T helper cell compartment. (**D**) Gating strategy of regulatory T helper cell subpopulations (Treg) in SFl and PBp after staining with panel 3. (**E**) Percentage of Treg as well as the proportion of CD45RA+ (naïve) Treg within the T helper cell compartment. (**F**) Percentage of CD45RA+ (naïve) T helper cells; * *p* < 0.05, ** *p* < 0.01, *** *p* < 0.001, **** *p* < 0.0001.

**Figure 4 ijms-23-04848-f004:**
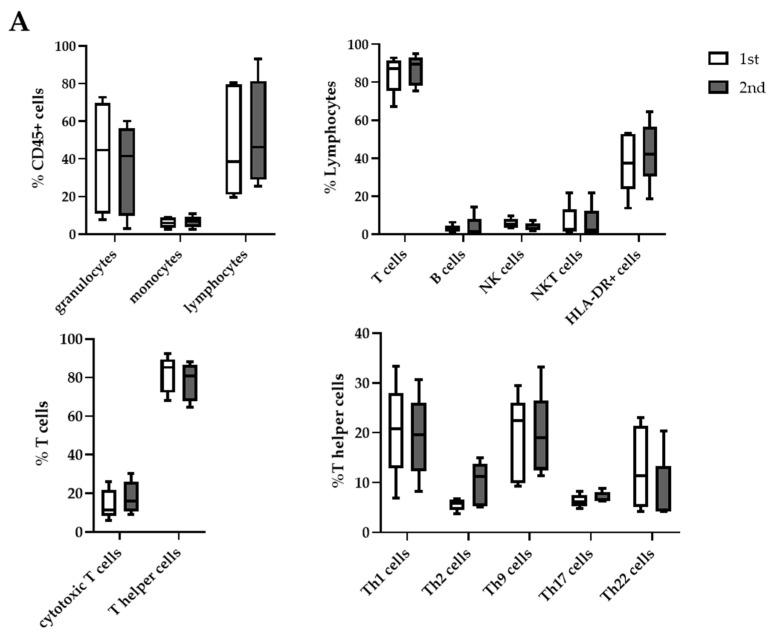
Comparison of all leucocyte subpopulations between first and second fine-needle aspiration.

**Table 1 ijms-23-04848-t001:** Description of tumor characteristics of 16 patients and 18 tumors (2 contralateral).

Tumor Characteristics	Subgroups	n (Number)
Histopathological type		
	NST Invasive lobular mucinous apocrine only DCIS solid papillary	11 3 1 1 1 1
Focality (2 bilateral carcinomas)		
	unifocal multifocal multicentric	12 2 4
Hormone receptor status (without DCIS, 2 bilateral carcinomas)		
	HR + ER-/PR-	14 3
Her-2/neu (without DCIS; 2 bilateral carcinomas)		
	positive negative	2 15
Ki67 (without DCIS; 2 bilateral carcinomas)		
	<20% ≥20%	8 9
Tumor size (2 bilateral carcinomas)		
	ypT0 pTis (y)pT1a (y)pT1b (y)pT1c (y)pT2 (y)pT3 (y)pT4	3 1 1 2 2 5 4 0
Axillary nodal status (2 bilateral carcinomas)		
	(y)pN0 (y)pN1 (y)pN2	12 3 2
Grading (without DCIS; 2 bilateral carcinomas)		
	G1 G2 G3	2 11 4

**Table 2 ijms-23-04848-t002:** Description of antibody panels used for flow cytometry analysis.

Panel 1 (all Beckman Coulter)	Fluorescent Label
anti-human CD45 antibody	Chromium Orange
anti-human CD3 antibody	APC-AF750
anti-human CD19 antibody	ECD
anti-human CD4 antibody	APC
anti-human CD8 antibody	Pacific Blue
anti-human CD16 antibody	PE
anti-human CD56 antibody	PE
anti-human HLA-DR antibody	PC7
**Panel 2** (all Miltenyi Biotec)	
anti-human CD4 antibody	VioBlue
anti-human CD183 (CXCR3) antibody	VioBright FITC
anti-human CD194 (CXCR4) antibody	PE-Vio 770
anti-human CD196 (CCR6) antibody	PE
anti-human CCR10 antibody	APC
**Panel 1** (all Beckman Coulter)	
anti-human CD45 antibody	Chromium Orange
anti-human CD3 antibody	APC-AF750
anti-human CD4 antibody	APC
anti-human CD127 antibody	FITC
anti-human CD25 antibody	PE
anti-human CD45RA antibody	ECD

## Data Availability

Data are available upon request from the corresponding author due to ethical reasons.

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
