# Peer review of "Seroma after Simple Mastectomy in Breast Cancer—The Role of CD4+ T Helper Cells and the Evidence as a Possible Specific Immune Process"

_ijms, 2022, doi:10.3390/ijms23094848_

Round 1

Reviewer 1 Report

Present manuscript by Nicole et al attempts to characterize cells in seroma after mastectomy and shows the presence of various CD4+T cell subsets in the site. Manuscript adds useful information to the existing literature but needs a thorough revision, in order to make an impact and understanding better. Methods are largely performed well but need mentioning of the details and demonstration of flow cytograms, mention about the gating strategies and staining procedures. I recommend a thorough revision.

Here are some major comments:

  1. Figure Legends should come at the end. Different parts of each figure should be merged together and then followed by legend.
  2. Gating strategy used throughout the manuscript should be included as supplementary figure. This should also include isotype/FMO/gating controls.
  3. A representative flow cytogram should always be included before showing the statistics so that reader can imagine what he/she is looking at.
  4. For Fig.1b, Do authors means events in the lymphocyte gate or everything recorded? It should be from a gated population because debris can have an influence otherwise. Regions should be marked around all populations
  5. Line 180: This link shows a 404 error. The method should be mentioned in the appropriate section and reference made to the company if needed.
  6. 3B the comparisons look reasonable but would make more trust if there are flow cytograms showing these populations.
  7. Line 200: Authors need to show flowcytograms and also whether they gated CD45RA vs CD45Ro or they are assuming CD45RA+ cells to be naive?
  8. Legend Fig.3D: This comparison should be split into the subsets that authors studied in previous figure
  9. Line 213: Do it means aspirations at two separate times? If so, the rationale and amount of time difference should be mentioned.
  10. Discussion seems very lengthy and has several things that can be excluded or shortened. It is reasonable to make things easy to understand but it should have a balance. Lines 222-230 for example are redundant and should be shortened. Line 244-245 can be removed or shortened.
  11. English language and some sentences need to be checked: for example line 220 nomination and differentiation should be changed to immunophenotyping or characterization.
  12. Line 235 cites two references but says ‘a publication’
  13. Line 241: Authors didn't investigate the monocyte subset to any depth... so that should rather be mentioned as a shortcoming of the study
  14. Line 242: It doesn't really looks so clear... there are infiltrating T cells, but same thing is also true for other immune cell subsets including granulocytes and monocytes.
  15. Line 254: Phrase specific response sounds like the triggers are known. Should be changed. Lines 263-265, make it difficult to understand because sentences are incomplete. If authors want to convey that presence of HLA-DR expressing cells at site of inflammation predicts an active immune response it should be stated clearly.
  16. Line 267: Age is a confounder and that would make comparison of blood less meaningful, but the comparison of SFl vs PBp are still very relevant and the basis for this study. This should be mentioned.
  17. Flow cytometry method needs to be mentioned in detail.

Author Response

Present manuscript by Nicole et al attempts to characterize cells in seroma after mastectomy and shows the presence of various CD4+T cell subsets in the site. Manuscript adds useful information to the existing literature but needs a thorough revision, in order to make an impact and understanding better. Methods are largely performed well but need mentioning of the details and demonstration of flow cytograms, mention about the gating strategies and staining procedures. I recommend a thorough revision.

Here are some major comments:

  1. Figure Legends should come at the end. Different parts of each figure should be merged together and then followed by legend.
    • Was changed the figures and legends according to reviewers suggestion.

  1. Gating strategy used throughout the manuscript should be included as supplementary figure. This should also include isotype/FMO/gating controls.
    • Gating strategies were added to all figures as well as suppl. Fig. 1A. FMO and Isotype controls were included (suppl. Fig 1 B / C) for panel 2. Panel 1 and 3 are part of our immune diagnostic unit, meaning they fulfill all requirements regarding GLP and quality management. Staining with these panels are performed using CE certified antibodies. Panels and gating strategy were establish by application specialists of Beckman Coulter. All stainings are controlled frequently, internally with reference material as well as externally by participating in inter-laboratory tests.

  1. A representative flow cytogram should always be included before showing the statistics so that reader can imagine what he/she is looking at.
    • Was changed according to reviewers suggestion and added flow cytograms to all figures.

  1. For Fig.1b, Do authors means events in the lymphocyte gate or everything recorded? It should be from a gated population because debris can have an influence otherwise. Regions should be marked around all populations
    • Figure was changed and gating strategy was added (Fig. 1C) as well as a description.

“Debris was exclude by size discrimination.”

  1. Line 180: This link shows a 404 error. The method should be mentioned in the appropriate section and reference made to the company if needed.
    • The link was deleted and changed into a reference (28).

  1. 3B the comparisons look reasonable but would make more trust if there are flow cytograms showing these populations.
    • We considered this comment carefully and added the information to suppl. Fig. 1A.

  1. Line 200: Authors need to show flowcytograms and also whether they gated CD45RA vs CD45Ro or they are assuming CD45RA+ cells to be naive?
    • Was changed according to reviewers suggestion and the gating strategy was added in Fig. 3D. Since we did not stain for CD45RO we assume that all CD45RA positive cells are CD45RO negative and thereby naïve.

  1. Legend Fig.3D: This comparison should be split into the subsets that authors studied in previous figure
    • Since the CD45RA antibody was not included in panel 2 (only in panel 3) it is not possible with those staining to show naïve cells within the different T helper cell subpopulations, only in the whole CD4+ T helper cell compartment.

  1. Line 213: Do it means aspirations at two separate times? If so, the rationale and amount of time difference should be mentioned.
    • Yes, seroma fluid aspirations were performed at two different time points. Additional information (e.g. time difference) was added in suppl. Table 3.

  1. Discussion seems very lengthy and has several things that can be excluded or shortened. It is reasonable to make things easy to understand but it should have a balance. Lines 222-230 for example are redundant and should be shortened. Line 244-245 can be removed or shortened.
    • Reviewers suggestion was considered and sentences were shortened or removed.

  1. English language and some sentences need to be checked: for example line 220 nomination and differentiation should be changed to immunophenotyping or characterization.
    • Reviewers suggestion was considered and changes were made.

  1. Line 235 cites two references but says ‘a publication’
    • Was changed according to reviewers suggestion, one reference was removed.

  1. Line 241: Authors didn't investigate the monocyte subset to any depth... so that should rather be mentioned as a shortcoming of the study
    • We considered this comment carefully and rewrote this part of the discussion as well as an addition to the results paragraph 2.2..

 Disscusion:

“In agreement with our findings, the authors showed an increased percentage of lymphocytes, as well as lower numbers of granulocytes and monocytes. This indicates a specific adaptive immune response leading to recruitment of specific lymphocytes to the side of injury. Instead of monocytes, Monalto et al. found cells with more heterogeneous physical characteristics and only partially expressing CD14. Since we did not stain for any monocyte markers, we cannot proof the latter statement. Nevertheless, we also saw those more heterogeneous population in the SSC vs. FSC plot. These cells might be particularly interesting for further investigations.”

Result:

“It is important to note, that granulocytes as well as monocytes in SFl did show a rather heterogeneous character and not so clear cells populations as compared to PBp/c in the SSC vs. FSC plot. This prompted us to draw gates for granulocytes as well as monocytes slightly bigger to be able to include all cells of the regarding cells type.”

  1. Line 242: It doesn't really looks so clear... there are infiltrating T cells, but same thing is also true for other immune cell subsets including granulocytes and monocytes.
    • We only state an “indication” of a specific immune response, since the other subsets like monocytes and granulocytes are very low in numbers and based on the SCC/FCC no clear cell populations. Nevertheless, to make an absolute statement a further characterisation of these cells is needed.

  1. Line 254: Phrase specific response sounds like the triggers are known. Should be changed. Lines 263-265, make it difficult to understand because sentences are incomplete. If authors want to convey that presence of HLA-DR expressing cells at site of inflammation predicts an active immune response it should be stated clearly.
    • Was changed according to reviewers suggestion.

“The significantly lower fraction of naïve CD4+ Th cells in blood of the patient cohort compared to healthy individuals could potentially be a biomarker for assessing risk of patients for seroma development.”

  1. Line 267: Age is a confounder and that would make comparison of blood less meaningful, but the comparison of SFl vs PBp are still very relevant and the basis for this study. This should be mentioned.
    • Was changed according to reviewers suggestion.

“However, SFl and PBp are taken form the study participant at the same time point which was the basement of this study.”

  1. Flow cytometry method needs to be mentioned in detail.
    • Addition of gating strategies were made to all figures and in paragraph 4.3..

Reviewer 2 Report

Pochert et al. performed an investigation of lymphocytes in seroma fluids after breast cancer surgery. The pilot study addresses an important complication. However, the manuscript has major flaws.

Major issues:

  • Please add descriptive statistical data concerning the timepoints of aspiration after surgery and the amount of seroma liquid (e.g. by adding the single data points in suppl table 3), including at least mean and median values.
  • Neoadjuvant / adjuvant chemotherapy and radiation might interfere with immune reactions and seroma formation. These factors should therefore be considered in a statistical analysis, e.g. is there a higher percentage of seroma in patients with chemotherapy or radiation, if the therapy precedes the seroma?
  • Please add a further control group of blood samples with surgery but without seroma formation. It has been mentioned in the conclusions part for further studies, however, it is a lacking control in the actual work.
  • Lines 265-267: “HLA-DR and CD45RA had been used as surrogate marker of immune competence in various clinical studies and its loss in circulation has been linked to post-surgical infections and sepsis.” This sentence is a verbatim copy of the introduction part of  the cited source 37 (Dunne et al). Moreover, sepsis has not been studied in the work of Dunne et al.

Minor issues:

  • Lines 38, 39: Please check / correct citation 5
  • Lines 39-41: Please check / correct citation 6
  • Lines 44-48: Please check / correct the sentence and citation 7
  • Lines 48, 49: Please check / correct citations 8, 9
  • Lines 49-51: Please check / correct citation 10-12. Further, the data of citation 12 neither supports the authors’ own postulations, nor your hypotheses.
  • Lines 51-53: Please check citation 5
  • Lines 71-73: T follicular regulatory cells are an established subtype, so there is more than one. However, whether more than two subtypes exist is an ongoing discussion. The tenor of the sentence might be ok, but it should not be too categorical.
  • Line 122: Please check / correct the sentence (grammar?): The gating strategy of panel 1was found (is displayed?) in Fig 1C.
  • Line 157: Please check / correct citation 28: as the paper deals with reference values of children, the oldest age group (16+) might not be the right to compare with.
  • Lines 177, 178: Please check / correct sentence.
  • Lines 238-245: The work of Montalto et al deals with another compartment, which might also be the cause of the difference. This should be considered as well.
  • Lines 259-261: The postulation is not covered by the data as healthy controls were used and not patients without seroma after surgery.
  • Lines 277-279: “Th22 cells are believed to have important roles both in promoting wound healing and repair of damaged epithelial barriers as well as in enhancing immune responses against some pathogens” -> This sentence is fairly close to the one of Gregor et al in https://www.sciencedirect.com/topics/immunology-and-microbiology/th22-cell (accessed 1 Apr 2022)
  • Table 1: “2 bilateral carcinoma” -> 2 bilateral carcinomas
  • Figure 1: Please add absolute numbers or number of cells per suitable volume unit in the figure legend of 1b and 1d.
  • Figure 2, 3: Please add absolute numbers or number of cells per suitable volume unit in the figure legends.

Please check / correct punctuation throughout the manuscript (e.g. p<0,05 -> p<0.05).

Author Response

Pochert et al. performed an investigation of lymphocytes in seroma fluids after breast cancer surgery. The pilot study addresses an important complication. However, the manuscript has major flaws.

Major issues:

  • Please add descriptive statistical data concerning the timepoints of aspiration after surgery and the amount of seroma liquid (e.g. by adding the single data points in suppl table 3), including at least mean and median values.
    • We considered this comment carefully and added the information to suppl. Table 3.

  • Neoadjuvant / adjuvant chemotherapy and radiation might interfere with immune reactions and seroma formation. These factors should therefore be considered in a statistical analysis, e.g. is there a higher percentage of seroma in patients with chemotherapy or radiation, if the therapy precedes the seroma?
    • We considered this comment carefully and added treatment approaches to the results paragraph 2.1. as well as suppl. Table 2.

“Suppl. Table 2 and 3 include information about the therapeutic approaches of the study group as well as the seroma aspirations including time points and volume of seroma fluid (SFl). Patients who received chemotherapy before surgery and an axilla dissection tending to develop more seromas. Patients who received chemotherapy before surgery develop seromas at equal frequencies compared to patients without chemotherapy. Moreover, the development of seromas with more than 100 ml volume seem to be not influenced by the type of surgery in the axilla.”

  • Please add a further control group of blood samples with surgery but without seroma formation. It has been mentioned in the conclusions part for further studies, however, it is a lacking control in the actual work.
    • We know that this control group is missing in this study. However, the aim of the study is to describe the cellular composition of seroma fluid in detail. As comparison, peripheral blood was used. To compare patients who undergo breast cancer surgery with and without seroma development an additional evaluation with a large cohort of serum samples is planned. Nevertheless, we have added preliminary data, which we now show in suppl. Fig 2. As expected, there are no significant differences between PBp with or without seroma formation. This is probably due to the small sample size or there are no measurable differences of the cell populations between patients with or without seroma development.

  • Lines 265-267: “HLA-DR and CD45RA had been used as surrogate marker of immune competence in various clinical studies and its loss in circulation has been linked to post-surgical infections and sepsis.” This sentence is a verbatim copy of the introduction part of the cited source 37 (Dunne et al). Moreover, sepsis has not been studied in the work of Dunne et al.
    • We considered this comment carefully and rewrote this part of the discussion.

“HLA-DR+ T helper cells were shown to be increased in patients with an active tuberculosis, furthermore Tippalagama et al. demonstrated that proliferating CD4+ effector T cells express high levels of HLA-DR. These findings suggest that HLA-DR might be a useful marker for monitoring immune responses.”

Minor issues:

  • Lines 38, 39: Please check / correct citation 5
    • This citation is correct.

  • Lines 39-41: Please check / correct citation 6
    • This citation is correct.

  • Lines 44-48: Please check / correct the sentence and citation 7
    • We v´changed the sentence to: There is evidence that the surgical trauma results in tissue damage, destroyed lymph vessels, invisible cell debris and fatty deposits as well as it results in the formation of a geometric dead space complex.
  • Lines 48, 49: Please check / correct citations 8, 9
    • These citation are correct.

  • Lines 49-51: Please check / correct citation 10-12. Further, the data of citation 12 neither supports the authors’ own postulations, nor your hypotheses.
    • Citation 12 was deleted. Citations 10 and 11 are correct.

  • Lines 51-53: Please check citation 5
    • Citation 5 was misplaced here and subsequently removed. The statement belongs to citation 13.

  • Lines 71-73: T follicular regulatory cells are an established subtype, so there is more than one. However, whether more than two subtypes exist is an ongoing discussion. The tenor of the sentence might be ok, but it should not be too categorical.
  • We considered this comment carefully and changed this part according to reviewers suggestion.

“T helper cells are divided into regulatory T cells (Treg) and effector T cells. These effector T cells can be subdivided into several types according to their cytokine secretion profile.”

  • Line 122: Please check / correct the sentence (grammar?): The gating strategy of panel 1was found (is displayed?) in Fig 1C.
  • Was changed the wording into “is displayed” according to reviewers suggestion.

  • Line 157: Please check / correct citation 28: as the paper deals with reference values of children, the oldest age group (16+) might not be the right to compare with.
    • Most publications about reference values state over 16 years old subjects as adults and do not further unravel other age groups. We used these reference values in particular, since those are obtained similarly to our gating strategy. We use the staining and gating method of panel 1 and those reference values in our immune diagnostics unit. All those measurements are controlled frequently, internally with reference material as well as externally by participating in interlaboratory tests.

  • Lines 177, 178: Please check / correct sentence.
    • We corrected the sentence according to the reviewers suggestion.

“To analyze effector T helper cell subpopulations, we adapted the gating strategy as described by an isolation protocol provided by Miltenyi Biotec (panel 2, Table 2) (Miltenyi Biotec, Bergisch Gladbach, Germany). We were able to determine the Th1, 2, 9, 17, and 22 subpopulations within the CD4+ T helper cell compartment for SFl as well as PBp/c respectively based on their lineage specific chemokine receptor profiles (Fig 3C and suppl. Figure 1).”

  • Lines 238-245: The work of Montalto et al deals with another compartment, which might also be the cause of the difference. This should be considered as well.
  • We considered this comment carefully and rewrote this part of the discussion as well as an addition to the results paragraph 2.2..

Disscusion:

“In agreement with our findings, the authors showed an increased percentage of lymphocytes, as well as lower numbers of granulocytes and monocytes. This indicates a specific adaptive immune response leading to recruitment of specific lymphocytes to the side of injury. Instead of monocytes, Monalto et al. found cells with more heterogeneous physical characteristics and only partially expressing CD14. Since we did not stain for any monocyte markers, we cannot proof the latter statement. Nevertheless, we also saw those more heterogeneous population in the SSC vs. FSC plot. These cells might be particularly interesting for further investigations.”

Result:

“It is important to note, that granulocytes as well as monocytes in SFl did show a rather heterogeneous character and not so clear cells populations as compared to PBp/c in the SSC vs. FSC plot. This prompted us to draw gates for granulocytes as well as monocytes slightly bigger to be able to include all cells of the regarding cells type.”

  • Lines 259-261: The postulation is not covered by the data as healthy controls were used and not patients without seroma after surgery.
  • We considered this comment carefully and rewrote this part of the discussion. “The significantly lower fraction of naïve CD4+ Th cells in blood of the patient cohort compared to healthy individuals could potentially be a biomarker for assessing risk of patients for seroma development.”

  • Lines 277-279: “Th22 cells are believed to have important roles both in promoting wound healing and repair of damaged epithelial barriers as well as in enhancing immune responses against some pathogens” -> This sentence is fairly close to the one of Gregor et al in https://www.sciencedirect.com/topics/immunology-and-microbiology/th22-cell (accessed 1 Apr 2022)
  • We added Gregor et al. as a citation.

  • Table 1: “2 bilateral carcinoma” -> 2 bilateral carcinomas
  • Was changed wording according to reviewers suggestion.

  • Figure 1: Please add absolute numbers or number of cells per suitable volume unit in the figure legend of 1b and 1d.
  • Figure 2, 3: Please add absolute numbers or number of cells per suitable volume unit in the figure legends.
    • Since the data was acquired using flow cytometry without any types of counting beads, we were not able to annotate absolute cell numbers. In addition, we used staining protocol with multiple washing steps that prohibit us from using cell numbers. The statement of percentages is the correct unit to use in our case. However, we made sure to record an appropriate number of cells (50.000 per staining).
  • Please check / correct punctuation throughout the manuscript (e.g. p<0,05 -> p<0.05).
    • We checked the punctuation throughout the manuscript and corrected mistakes.

Round 2

Reviewer 2 Report

The work by Pochert et al. has been ameliorated in an appropriate manner in its revised version, the inclusion of a control group -yet small- is appreciated. According to the authors, the aim of the study is to describe the cellular composition of the seroma fluid in detail. Still, the title “The role of CD4+ T helper cells and the evidence as a possible specific immune process” is more than a weak immunological postulation, which was the reason to demand / include a group with an ongoing wound healing, which is an immunological process as well, that needs to be at least controlled in the data. The major issues have been appropriately addressed, I have only some minor issues left.

Minor issues:

  • Lines 38, 39: It was claimed in the rebuttal that citation 5 is correct. The sentence is as follows: “Comparing different studies in 3-85% of all cases 38 seroma formation occurs mostly within the first weeks after breast surgery 5.” At least the accessible part of citation 5 comprises one study “…Sixty-four consecutive female patients undergoing Halsted mastectomy were prospectively studied for this purpose. …” Historical articles as the cited one are not accessible online. If citation 5 is inspired by citation 6 at position [2] (“…Seroma formation is the most frequent postoperative complication seen after mastectomy and axillary surgery with an incidence of 3% to 85% [2]. …”), the authors should give a clear textual context justifying citation 5 e.g. by providing the part of the original article where this statement is made. If the authors do not have access to the fulltext article, they should rather consider to edit the sentence e.g. considering the work of Kuroi et al (PMID 16286909).
  • Lines 44-46: The points raised might be supported, but are not covered by the data of citation 7.
  • Lines 47-50: The origin of the seroma fluid was not studied in citations 10 or 11. The articles rather describe risk factors and possible treatment strategies. Consider citations 12, 13 instead.
  • Line 157: Please check / correct citation 28: as the paper deals with reference values of children, the oldest age group (16+) might not be the right to compare with. -> “Most publications about reference values state over 16 years old subjects as adults and do not further unravel other age groups. <<We used these reference values in particular, since those are obtained similarly to our gating strategy.>> We use the staining and gating method of panel 1 and those reference values in our immune diagnostics unit. All those measurements are controlled frequently, internally with reference material as well as externally by participating in interlaboratory tests.” I agree with your arguments concerning this issue, consider to include the highlighted part in the materials and methods.
  • Figure 1: Please add absolute numbers or number of cells per suitable volume unit in the figure legend of 1b and 1d. Figure 2, 3: Please add absolute numbers or number of cells per suitable volume unit in the figure legends. -> “Since the data was acquired using flow cytometry without any types of counting beads, we were not able to annotate absolute cell numbers. In addition, we used staining protocol with multiple washing steps that prohibit us from using cell numbers. The statement of percentages is the correct unit to use in our case. However, <<we made sure to record an appropriate number of cells (50.000 per staining).>>” There is no doubt about the correct use of percentages in your case, which is however based upon a certain number of cells, even though the ground truth is unknown. Please add the highlighted content in the materials and methods part 4.3.
  • Minor grammar and spell check required.

Author Response

The work by Pochert et al. has been ameliorated in an appropriate manner in its revised version, the inclusion of a control group -yet small- is appreciated. According to the authors, the aim of the study is to describe the cellular composition of the seroma fluid in detail. Still, the title “The role of CD4+ T helper cells and the evidence as a possible specific immune process” is more than a weak immunological postulation, which was the reason to demand / include a group with an ongoing wound healing, which is an immunological process as well, that needs to be at least controlled in the data. The major issues have been appropriately addressed, I have only some minor issues left.

Minor issues:

  • Lines 38, 39: It was claimed in the rebuttal that citation 5 is correct. The sentence is as follows: “Comparing different studies in 3-85% of all cases 38 seroma formation occurs mostly within the first weeks after breast surgery 5.” At least the accessible part of citation 5 comprises one study “…Sixty-four consecutive female patients undergoing Halsted mastectomy were prospectively studied for this purpose. …” Historical articles as the cited one are not accessible online. If citation 5 is inspired by citation 6 at position [2] (“…Seroma formation is the most frequent postoperative complication seen after mastectomy and axillary surgery with an incidence of 3% to 85% [2]. …”), the authors should give a clear textual context justifying citation 5 e.g. by providing the part of the original article where this statement is made. If the authors do not have access to the fulltext article, they should rather consider to edit the sentence e.g. considering the work of Kuroi et al (PMID 16286909).
    • We considered this comment carefully, rewrote the sentence, and edited the citations.

“In reviews comparing different studies in 3% respectively 10% up to 85% of all cases seroma formation occurs mostly within the first weeks after breast surgery 5, 6.”

  • Lines 44-46: The points raised might be supported, but are not covered by the data of citation 7.
    • We considered this comment carefully and rewrote the sentence.

“There is evidence that the surgical trauma after sentinel lymph node biopsy for breast cancer treatment and seroma development is associated with age at least in some seroma development studies 8.”

  • Lines 47-50: The origin of the seroma fluid was not studied in citations 10 or 11. The articles rather describe risk factors and possible treatment strategies. Consider citations 12, 13 instead.
    • We changed the citations according to reviewers suggestion.

  • Line 157: Please check / correct citation 28: as the paper deals with reference values of children, the oldest age group (16+) might not be the right to compare with. -> “Most publications about reference values state over 16 years old subjects as adults and do not further unravel other age groups. <<We used these reference values in particular, since those are obtained similarly to our gating strategy.>> We use the staining and gating method of panel 1 and those reference values in our immune diagnostics unit. All those measurements are controlled frequently, internally with reference material as well as externally by participating in interlaboratory tests.” I agree with your arguments concerning this issue, consider to include the highlighted part in the materials and methods.
    • We added this sentence to the paragraph 2.3..

  • Figure 1: Please add absolute numbers or number of cells per suitable volume unit in the figure legend of 1b and 1d. Figure 2, 3: Please add absolute numbers or number of cells per suitable volume unit in the figure legends. -> “Since the data was acquired using flow cytometry without any types of counting beads, we were not able to annotate absolute cell numbers. In addition, we used staining protocol with multiple washing steps that prohibit us from using cell numbers. The statement of percentages is the correct unit to use in our case. However, <<we made sure to record an appropriate number of cells (50.000 per staining).>>” There is no doubt about the correct use of percentages in your case, which is however based upon a certain number of cells, even though the ground truth is unknown. Please add the highlighted content in the materials and methods part 4.3.
    • We added this sentence to the paragraph 4.3..
  • Minor grammar and spell check required.
    • Grammar and spelling was checked and minor changes were made.
